# Experimental Validation of Diffraction Lithography for Fabrication of Solid Microneedles

**DOI:** 10.3390/ma15248934

**Published:** 2022-12-14

**Authors:** Jun Ying Tan, Yuankai Li, Faraz Chamani, Aabila Tharzeen, Punit Prakash, Balasubramaniam Natarajan, Rahul A. Sheth, Won Min Park, Albert Kim, Donghoon Yoon, Jungkwun Kim

**Affiliations:** 1Department of Electrical Engineering, University of North Texas, Denton, TX 76207, USA; 2Department of Electrical and Computer Engineering, Kansas State University, Manhattan, KS 66506, USA; 3Department of Interventional Radiology, University of Texas MD Anderson Cancer Center, Houston, TX 77030, USA; 4Tim Taylor Department of Chemical Engineering, Kansas State University, Manhattan, KS 66506, USA; 5Department of Medical Engineering, The University of South Florida, Tampa, FL 33620, USA; 6College of Medicine, University of Arkansas for Medical Science, Little Rock, AR 72205, USA

**Keywords:** solid microneedles, inclined microneedles, diffraction UV lithography, photosensitive resin, microlens, first harmonic, first bell-tip, second harmonic, microfabrication, skin insertion

## Abstract

Microneedles are highly sought after for medicinal and cosmetic applications. However, the current manufacturing process for microneedles remains complicated, hindering its applicability to a broader variety of applications. As diffraction lithography has been recently reported as a simple method for fabricating solid microneedles, this paper presents the experimental validation of the use of ultraviolet light diffraction to control the liquid-to-solid transition of photosensitive resin to define the microneedle shape. The shapes of the resultant microneedles were investigated utilizing the primary experimental parameters including the photopattern size, ultraviolet light intensity, and the exposure time. Our fabrication results indicated that the fabricated microneedles became taller and larger in general when the experimental parameters were increased. Additionally, our investigation revealed four unique crosslinked resin morphologies during the first growth of the microneedle: microlens, first harmonic, first bell-tip, and second harmonic shapes. Additionally, by tilting the light exposure direction, a novel inclined microneedle array was fabricated for the first time. The fabricated microneedles were characterized with skin insertion and force-displacement tests. This experimental study enables the shapes and mechanical properties of the microneedles to be predicted in advance for mass production and wide practical use for biomedical or cosmetic applications.

## 1. Introduction

Recently, medical therapy utilizing microneedles has laid great emphasis on developing the next generation of drug delivery methods. Hypodermic needle-based medication administration system has been historically employed in a broad range of medical practices, it has been known to have several downsides including high prevalence of needle phobia [1,2], the generation of considerable medical waste [3], and the requirement for an expensive cold chain supply [4,5]. While transdermal patch-based medication administration is tempting because it avoids the downsides associated with hypodermic needle delivery, several drawbacks of transdermal patches such as poor drug efficacy and restricted drug availability remain obstacles. These obstacles are mainly due to the outermost skin barrier, known as the stratum corneum, which functions as a protective barrier on human skin, preventing external substances, including toxins, bacteria, proteins, and macromolecules from entering the human body [6]. Oral administration allows for more regulated medication release over a longer period of time [7,8]. However, most macromolecules such as proteins and insulin restrict the use of oral options primarily due to their poor absorption and vulnerability to the degradation in the gastrointestinal tract and liver [9,10]. Conversely, microneedle-based devices that offer local administration and systematic and controlled delivery have emerged as an efficient drug delivery method for both small and macromolecules [11].

Microneedles are needles that are minuscule in size, generally ranging in length from 100 to 1500 µm. Conical-shaped microneedles with circular base, width of 300 µm and height of 600 µm have been the most commonly investigated microneedle type in recent years [11,12]. In 2015, Alkilani et al. reported that microneedle-integrated transdermal patches can produce micropores upon insertion, increasing skin permeability, and thereby allow for increased drug delivery [13]. Additionally, drug administration with microneedles only caused relatively small discomfort as compared to the conventional hypodermic needle injection. The visual analog scale (VAS) is a frequently used subjective method for determining pain levels. Across all trials that evaluated the amount of pain generated by microneedle insertion, the VAS score for the microneedle technique was much lower than for the traditional hypodermic needle approach, regardless of the diameter, shape, or number of microneedles [14,15,16]. Additionally, microneedle patches and devices have been reported to be safe for self-administration without the assistance of fully trained healthcare workers [17,18,19,20]. Microneedles puncture marks also heal more quickly than hypodermic needle puncture marks [21]. 

Throughout the decades since the microneedle was first used for drug delivery [22], the microfabrication methods for microneedles have been developed using both top-down methods including laser ablation micromachining [23,24,25,26] as well as bottom-up fabrication such as direct drawing lithography [27,28,29,30], three-dimensional (3D) printing [31,32,33,34], and two-photon polymerization [35,36,37]. The top-down technique was historically a serial process, making it unsuitable for batch manufacturing and necessitating the purchase of expensive equipment. Bottom-up techniques are commonly utilized to manufacture microneedles layer by layer via photopolymerization, which previously resulted in poor needle body surface quality, poor needle tip diameter (sharpness), and lengthy process times due to the high number of procedures. Recently, a low-cost, quick UV lithography technology for microneedle production was presented. In 2017, Takahashi et al. demonstrated a quick and simple microneedle fabrication method via UV lithography process with rotating prism [38]. In 2021, Lim et al. presented a microneedle mold fabrication method via inclined and rotating UV exposure process, which can be later replicated with poly(lactic-co-glycolic) acid (PLGA) to form the microneedles [39]. Both of these approaches have demonstrated microneedle with smooth surface, however, the requirement for prism, rotatable UV exposure system and a lengthy baking time complicates the process and hinder the feasibility for mass production. Another notable microneedle fabrication method published by Dardano et al. in 2021, which incorporated a direct UV exposure approach has proven feasible for large scale manufacturing, however, the fabricated microneedles were inconsistent with poor surface quality [40]. 

After demonstrating a way for forming a few millimeter tall microneedle shape using direct UV exposure to a liquid state photosensitive photopolymer [41], we studied the liquid to solid transition using UV diffraction as a microneedle production method [42,43,44]. The microneedle production approach based on diffraction lithography introduced self-aligned light focused on producing the microneedles and was demonstrated to be a simple, low-cost, and scalable fabrication process. The key characteristics of this technology demonstrated its simplicity in producing discrete microneedle with various heights and kinds on the same substrate concurrently. In the present article, we examined the relationship between the applied UV radiation and the morphology of the microneedles in further detail. We observed a series of variations in the light propagation occurring consistently during the process, which was caused by the photosensitive resin’s liquid to solid transition. Throughout an extensive examination, we recognized four significant phases of the solidified structure during the transition, which include the formation of a microlens, first harmonic, first bell-tip, and second harmonic as depicted in Figure 1. Each phase was named after the signature features of the solidified structure at the current phase. For instance, microlens phase occurs when the liquid resin first transforms into a solid form, it presents as a plano-convex shape, and thus, it was named as microlens phase. First harmonic phase occurs when the microlens develops into a conical shape with an acute tip at the vertex. First bell-tip phase appears when the acute tip is elongated vertically and forms a bell-tip at the vertex, which follows by the second harmonic phase that features a horizontally broadened bell-tip. Additionally, we describe the inclined microneedle for the first time, as depicted in Figure 1. Given that the human skin is the biggest organ on the body, several applications using microneedles, such as biosensors and therapy devices, are envisaged.

## 2. UV Light Propagation

The geometry of the light energy distribution is determined by the UV light propagating via the photomask patterns to the photosensitive resin. The microneedle’s solid form is determined by the border of the dispersed light energy between crosslinking and non-crosslinking of the photosensitive resin. Since the area of the threshold energy required to crosslink the photosensitive resin is time-dependent, assuming constant input light intensity, the route of light propagation varies correspondingly. 

### 2.1. Experimental Setup for UV Light Visualization

To verify the alterations in the light route caused by resin crosslinking, the real UV light propagation inside the photosensitive resin was viewed under a microscope, as shown in Figure 2. The images were captured using a digital optical microscope (Smartzoom 5, Zeiss Inc., Oberkochen, Germany). The magnification was kept constant, but the focus of the microscope was adjusted slightly to track the growth of the microneedle. For clearer imaging, a grey resin (FLGPGR04, Formlabs, Inc., Somerville, MA, USA) was used as the grey parameter slightly scatters the light that can be observed under the microscope. However, the clear surgical guide resin (FLSGAM01, Formlabs, Inc., Somerville, MA, USA) was used for the rest of the structural demonstration as it is known as biocompatible and simple in fabrication [45]. Due to the high attenuation grey surgical guide resin has in comparison to clear surgical guide resin, the light intensity utilized in this experiment was about three times more than the intensity needed to crosslink clear surgical guide resin to get the same outcome. 

### 2.2. COMSOL Multiphysics Simulation Condition for UV Light Propagation

Furthermore, the experimental visualized UV light propagation was also justified with simulations using the optics module of COMSOL Multiphysics^®^ Software (COMSOL Inc., Stockholm, Sweden). The simulation scheme was modeled with two 110 nm thick chromium layers sited on a glass substrate with a gap of 150 µm between them to simulate the photomask, and a rectangular block with sufficient thickness sited on top of the photomask to simulate the coated resin or air. A plane wave with wavelength of 405 nm was assigned to propagate in the +Z direction from the bottom edge to the glass substrate. All outermost boundaries were modeled to allow the incoming light to exit freely to avoid internal reflection. The refractive index of air, glass, liquid resin and solid resin were set to be 1, 1.5, 1.46 and 1.5403, and the attenuation factor of the resins was set to be 0.002878, these values were obtained from the manufacturer and previous experiments [43,46,47]. The maximum mesh element size was set to be λ/5. 

### 2.3. Experimental Visualization Results of UV Light Propagation

150 µm diameter circular photo patterns in 4 by 4 array were used as substrate. Collimated UV light with a wavelength of 405 nm was exposed from the back of the photomask onto the photosensitive resin. The UV light propagation through the photopatterns was captured at the beginning of the light exposure in Figure 2a, where the original photopatterns were revealed with blue UV light. Figure 2b–e demonstrate the progress of the actual light propagation associated with crosslinking progress of the resin. As an initial distinguishable crosslinked shape of the resin, a plano-convex microlens shape was found in a few seconds of exposure. Since the refractive index of the crosslinked resin is higher than that of the liquid state resin, the microlens shape solid structure started to converge towards a focal point as demonstrated in Figure 2b. As the UV exposure continued, the resin was crosslinked and solidified through the light path to form the first conical shape of the microneedle, indicated as the first harmonic (Figure 2c). Incident UV light that travels through the center of the photopatterns exited the vertex vertically, whereas the rest of the incident UV light that was guided by the microneedle body and diverged at the vertex of the first harmonic as shown in Figure 2c, the resultant of this UV light propagation formed the first bell-tip (Figure 2d). Figure 2d reveals that as the light exited the first bell tip, it diverged further whilst the strong beam at the center continued to solidify the resin and extended vertically, which eventually formed the second harmonic microneedle (Figure 2e). Based on the observation, the UV exposure first formed the microlens shape that redirected the light propagation to focus down. Further applied UV light forms the first harmonic of the microneedle. The continued light application caused the light scattering at the vertex of the first cone to form the unique first bell-tip of the microneedle. By applying more exposure energy, the first bell-tip extends vertically and broadens horizontally, which forms the second harmonic of the microneedle. By applying more UV light, we observed up to fourth harmonic structures. It assumes that the solidified harmonic structures work as a light waveguide to send the light to crosslink more harmonic structures. When the collimated UV light is introduced at an angle, inclined microlens, first harmonic, first bell-tip, and second harmonic of the microneedle can be achieved based on the same principles.

### 2.4. Simulation Results of UV Light Propagation

Figure 2g–j demonstrates a side-by-side comparison of the visualized and simulated UV light propagation to better understand the alteration of the UV light propagation at the four transition phases. Figure 2f shows the simulated UV light propagation in air, since there is no change in the property of the medium, thus, the UV light propagation remained undistorted. Figure 2g shows the simulated UV light propagation after the formation of a plano-convex microlens, where light converged towards a focal point, suggesting that the microlens will rapidly grow vertically towards a point as guided by the UV light propagation and eventually transform into a conical shape, known as first harmonic. The simulated result also corresponds to the actual visualized UV light propagation under microscope. Figure 2h depicts the simulated UV light propagation once the first harmonic has formed. During this phase, light no longer converges but diverges as it exits the vertex of the first harmonic. The simulated light implies that the first harmonic’s vertex will grow in both vertical and horizontal directions, resulting in an extended and broader tip and entering the first bell-tip phase. The simulation result in Figure 2i shows the UV light propagation in the first bell-tip phase, where the majority of light diverged at the side first bell-tip while the minority exited vertically, indicating that a slowdown in vertical growth and tendency to broaden, resulting in a slightly taller but significantly wider bell-tip, known as the second harmonic phase. In the second harmonic phase as shown in Figure 2j, the body of second harmonic guides the light towards its vertex and exits with a diverging pattern similar to Figure 2h. This simulation result suggests that a repeating cycle beyond second harmonic will occur as the exposure continues until the light has completely dispersed by attenuation.

## 3. Fabrication

The fabrication process of both straight and inclined microneedles is depicted in Figure 3. A commercial liquid-state photosensitive resin (Surgical guide resin, Formlabs, Inc.) was coated on a photomask with circular photopatterns (Figure 3a). The soda lime photomask was purchased from Telic company and micro-patterned using a maskless projection lithography mask writer (SF-100 XPRESS, Scotech Inc., Tokyo, Japan). The resin is poured on the photomask/substrate sufficiently. Unlike the conventional photoresist coating, no precision control of the resin thickness is required, nor is the baking process needed. Collimated UV light sourced from a narrow-band UV-LED (λ = 405 nm; Shenzhen Chanzon Technology Co., Ltd., Shenzhen, China) was introduced from the backside of the glass substrate for UV exposure (Figure 3b). The light collimation from the LED was customized by introducing a simple plano-convex lens with a 3D printed waveguide [48]. The UV exposure was applied until the solidified resin had reached or formed the desired height or needle type. The sample was developed in isopropanol for 10 min to remove all the remaining liquid resin (Figure 3c). Slight orbital agitation (20 rpm) was introduced to ensure constant fluidic flow and prevent a stagnant environment. Lastly, the sample was dried with compressed air then treated with UV light for biocompatibility to complete the microneedles (Figure 3d). The inclined microneedles were achieved by introducing the collimated UV light at an angle then proceeding with the same remaining process.

## 4. Results and Discussion

The shape of the fabricated microneedle is affected by light diffraction parameters including the size of the photomask patterns, the light intensity, and the exposure time. These optical parameters determine the key characteristics of the microneedle including height, shape, and the base to the height aspect ratio, which are closely related to the microneedle’s functionality such as the skin insertion or the mechanical stability. This study demonstrates the correlation of each light diffraction parameter in associated with the key characteristics of the microneedle.

### 4.1. Photomask Patterns Size

In the first study, the effect of photomask pattern size on the fabricated microneedle was examined. 11 circular photomask patterns array with the size of 50, 75, 100, 125, 150, 175, 200, 250, 300, 400, and 500 µm were prepared on the same photomask substrate using standard photolithography and wet etching process. All photomask patterns were positioned side-by-side with 300 µm spaced in between to ensure no interference between the adjacent patterns. The photosensitive resin was directly applied to the photopatterns as a backside UV exposure scheme where the amount of the photosensitive resin was thicker than the height of the targeted microneedle. The intensity of the UV light from the UVLED (405 nm) was fixed at 2 mW/cm^2^, and the UV exposure time was set at 150 s resulting in the total applied UV exposure energy of 300 mJ/cm^2^. The collimated UV light was applied from the backside of the photomask. The fabrication results show that the height of the microneedles formed around 2 to 4 times longer in general than the diameter of the photopattern, as shown in Figure 4a. The tallest fabricated microneedle in this experiment was formed by the 500 µm pattern with the measured height of 1020 µm, whereas the shortest was formed by 50 µm pattern with height of 160 µm. From the experimental results, we observed a good correlation between the size of the photomask pattern and the height of the microneedles, where the height of the microneedle increases as the pattern size increases. Besides the microneedle height, the shape of the microneedles was also affected by the photomask pattern size. Given the same intensity and exposure time, 50 µm pattern has formed the microneedle with bell-tip while 500 µm pattern has formed only the first harmonic. Other examples such as microneedles with pattern size of 250 µm and 300 µm shows an intermediate microneedle shape between first harmonic and first bell-tip phases. Based on this result, we observed that photomask pattern with smaller size progresses faster through each microneedle phase, whereas larger photomask pattern size progresses slower. To summarize the first study, photomask pattern with larger size forms taller microneedle but shows slow progression through the microneedle phases, whereas photomask pattern with smaller size forms shorter microneedle but progresses faster through the microneedle phases, and thus, the height and the shape of the microneedle can be predicted with various photopattern sizes with the constant UV light intensity and the exposure time.

### 4.2. UV Light Intensity

The study of the UV light intensity effect on the fabricated microneedles was followed to observe. In this experiment, the photomask pattern of diameter (150 µm) and exposure time (30 s) were used as control, while various intensities of UV light were applied including 1, 1.5, 2, 4, 10, 20, 40, 60, 80 and 90 mW/cm^2^, where 90 mW/cm^2^ was the maximum intensity allowed by the UV exposure system. Figure 4b shows the experimental result, where the higher the UV light intensity, the taller the microneedle was formed. The result was expectable since light with higher intensity typically travels farther distance while light with lower intensity diminished after short distance of travel due to diffraction. However, another aspect of this experiment was to investigate the tip-to-body aspect ratio of the fabricated microneedles. In the experimental result, intensities that range from 4 mW/cm^2^ to 90 mW/cm^2^ have formed microneedle with bell-tip. The microneedle that was formed with 4 mW/cm^2^ shows a tip-to-body aspect ratio of 0.26 where the microneedle that was formed with 90 mW/cm^2^ shows aspect ratio of 0.95. From this experiment, we have learned that the UV light intensity affects both the shape and the height of the microneedle. In brief, the tip-to-body aspect ratio of the microneedle increases as the UV light intensity increases given the constant photomask pattern diameter and exposure time. With the precise control of UV light intensity, all types of microneedles can be fabricated with the desired microneedle tip-to-body aspect ratio. 

### 4.3. UV Exposure Time

The third study revolves around the effect of the total exposure time on the height and shapes of the microneedle. The same photomask pattern size of the 150 µm diameter was used, and the UV light intensity was set at 2 mW/cm^2^ as control, while exposure time was set as the variable ranging from 10 to 1200 s. The result of this study demonstrated that longer exposure time caused the growing height as well as the different shapes of the microneedles as shown in Figure 4c. Four different noticeable shape changes were found. It started from a lens shape and followed by a conical shape named as a first harmonic. With further exposure, a bell-tip was found on top of the first harmonic and followed by a broadened, sharp oval shape, named as a second harmonic structure. In detail, the lens shape was formed at 10-s and grown up to the 20 s. After 5 s of the intermediate state, the microneedle first exhibited the first harmonic characteristics (sharp tip) at 25 s and remained in the same phase up to 30 s. Then, the microneedle progressed through a 90 s intermediate state, which the elongation of the tip was observed but remain unsharp. The bell-tip phase was first completed at 120 s and continued to elongate until 400 s. At 400 s, distinguishable broadened tip was observed, which implies the beginning of the second harmonic phase. There is no intermediate phase between the bell-tip and the second harmonic phase. The microneedle continued to elongate and broaden until 1200 s, which formed a unique second harmonic structure with tip-to-body aspect ratio of 1. From the presented experimental data, we observed the progression of four major phases of the microneedle within a single exposure. By changing the exposure time with constant photomask pattern size and UV light intensity, various heights and shapes of microneedle can be fabricated.

### 4.4. Refractive Index

In addition, we examined the refractive index of the resin that we used in the previous experiments. As the microneedle is formed from the solidification of the liquid state photosensitive resin, the refractive index of the resin can show the effect of the UV light incident angle on the inclined angle of the microneedle body. However, as the introduced UVLED is not a single wavelength light source, the refractive index of the resin corresponding to the specified UVLED needs to be experimentally verified. A 150 µm circular photopattern was used as the test subject and UV light with the intensity of 2 mW/cm^2^ was introduced at various angles (15°, 30°, 45°, and 60°) for 30 s of exposure time. Five batches of microneedles were produced and measured for each inclined angle, and the mean values of the measured microneedle inclined angle are 10.2°, 20.0°, 29.1°, and 36.4°. The refractive index of the resin could be estimated using the Snell-Descartes equation of refraction based on the results [49]:(1)$$n_{1}\sin\left(\theta_{i}\right)=n_{2}\sin\left(\theta_{r}\right)$$
where $n_{1}$ is the refractive index of air, $n_{2}$ is the refractive index of the resin, $\theta_{i}$ is the UV light incident angle and $\theta_{r}$ is the refracted light angle, which is also the inclined angle of the microneedle body. The refractive index of the resin was calculated to be 1.4607 from the experimental data. 

### 4.5. Large Scale Fabrication Results

To illustrate the batch process capabilities, a straight and inclined first bell-tip microneedles array, as shown in Figure 5, was fabricated. The microneedles arrays include 180 units of microneedles with a base diameter of 150 µm and a height of 460 ± 14 µm for straight microneedles (Figure 5a) and 450 ± 10 µm for inclined microneedles with a 20° inclined angle (Figure 5c). Both types of microneedles have a 3:1 aspect ratio, which was chosen to maintain the microneedle’s mechanical stability and avoid buckling. The tip size of the straight microneedles was determined to be 1.14 µm (Figure 5b), and the inclined microneedles were determined to be 1.4 µm (Figure 5d), which are regarded sharp enough (<10 µm) to pierce the skin. 

### 4.6. Skin Insertion Tests

A skin insertion test was performed to verify the microneedles array’s operation. A straight microneedle array with the first bell-tip depicted in Figure 5a was molded in polydimethylsiloxane (PDMS) and then transformed to a male microneedle array in polylactic acid (PLA). It should be mentioned that the resin-based microneedle was molded into a PLA microneedle to demonstrate the standard skin test [50], using commonly available material. This material conversion also expands our fabrication capabilities, since the microneedle material may be molded into a variety of materials. Pig cadaver skins were prepared from a grocery market for the insertion test. The PLA microneedles array was positioned downward on the pigskin, and thumb pressure was applied to the PLA substrate’s backside. After completely inserting the PLA microneedles array into the pigskin, the array was withdrawn, and the implanted region dyed with blue tissue marking dye (Tissue marking dye, Cancer Diagnostics Inc., Durham, NC, USA) for viewing. As illustrated in Figure 6a, the PLA microneedles were successfully implanted into the pigskin. A two-by-one inclined first harmonic microneedle array with a diameter of 300 µm and a height of 800 µm was produced and utilized for the skin insertion test. The successful insertion of the tips into the pigskin and detachment of the needle body from the glass substrate are seen in Figure 6b. 

### 4.7. Force-Displacement Test on Straight Microneedle

The PLA microneedles were subjected to a force-displacement test to ascertain their mechanical strength and behavior under compression. On an upward-facing platform, a single PLA straight first bell-tip microneedle with a diameter of 150 µm and a height of 550 µm was placed. On a motor-driven threaded rod that was controlled by a microprocessor (Arduino UNO Rev 3, Arduino, Turin, Italy), a force gauge (FC200, Torbal Inc., Clifton, NJ, USA) was incorporated. The motor was instructed to descend at a rate of 1.2 mm/min until the microneedle was completely squeezed. Figure 7 depicts the outcome of the force-displacement test. Figure 7b illustrates the tested microneedle prior to and after the compression. As the force gauge descended, a minor peak was observed at 61 mN, indicating the force necessary to shatter the microneedle tip. As compression proceeded, the compression force grew linearly until the force gauge released the pressure, at which point the highest force applied to the microneedle body was 1032 mN while the microneedle remained attached to the PLA substrate.

### 4.8. Force-Displacement Test on Inclined Microneedle

A force-displacement test was also conducted on resin-based inclined microneedles to investigate the mechanical strength and behavior under compression. A single resin-based inclined first bell-tip microneedles with a diameter of 150 µm, the height of 480 µm and an inclined angle of 20° were mounted on a platform facing upwards. The same force gauge was commanded to move downwards at a speed of 1.2 mm/min until the microneedle had been fully compressed. The result of the force-displacement test was presented in Figure 8a, where the conceptual drawing of the experiment setup was shown in the inset image. Figure 8b shows the zoomed in view of the force-displacement test result at the first 300 µm displacement, a small peak of 26.5 mN was measured, indicating the needle tip was broken. As the compression continued, the compression forces encountered a sudden drop at 1015 mN, which indicates the required amount of compression force to detach the microneedle from the substrate. The detached inclined microneedle was shown in Figure 8c. The force gauge continued to compress the body of the microneedle until 1319 mN before releasing the pressure. Another experiment was conducted to investigate the amount of force needed to detach the inclined microneedle when it was compressed at the same inclined direction (in-phase) or against the inclined direction (out-phase). The result of this experiment is shown in Figure 8d. When the inclined microneedle is compressed with in-phase forces, the detachment occurs when the force reaches 574 mN, as compared to out-phase compression force, the required force for detachment was only 224 mN. Based on the result of this experiment, it has been evidenced that inclined microneedle can be useful for applications that require microneedles to be broken after being inserted into the skin.

## 5. Conclusions

We demonstrated a novel fabrication method for microneedle based on UV light diffraction and photopolymerization. UV light transmitted via a microscale photopattern diffractionally generated a conical shape beneath the liquid photosensitive resin, which is referred to as a microneedle. During the liquid to the solid transition of the photosensitive resin, we found four distinct phases including microlens, first harmonic, first bell-tip and second harmonic structures. Those overall geometries are affected by three major optical parameters including UV light intensity, photopattern size, and exposure time. At the various photopattern designs with the constant light intensity and exposure, the height of the microneedles formed around 2 to 4 times longer in general than the diameter of the photopattern. At the various light intensity with constant photopattern size and exposure, the microneedle tip-to-body aspect ratio increases as the intensity increases. At the constant photopattern size and the light intensity, the microneedle progresses through four major phases with increasing exposure time. Additionally, the refractive index of the surgical guide resin was found to be 1.4607. The batch fabrication capability demonstrated 180 units of straight microneedles array with first bell-tip as well as 180 units of inclined microneedles array with first bell-tip. Skin insertion tests and force-displacement tests for both straight and inclined microneedles array proved their functionality as microneedles. The molding process with the fabricated resin-based microneedles demonstrated PLA-based straight microneedle and its good mechanical characteristics including a tip strength of 61 mN and body strength of 1032 mN. The inclined microneedle showed good stability with a tip strength of 26.5 mN and detachment force of 1015 mN under the vertical compression that is stronger tolerance against in-phase horizontal compression than out-phase. This inclined microneedle has considerable promise for applications that need microneedles to stay in the skin for a prolonged amount of time after the patch is removed. In general, the suggested fabrication approach involves just a single resin coating and a single UV direct exposure, which can be finished in less than 30 min and has significant industrial potential as well as applicability in biological applications.

## Figures and Tables

**Figure 1 materials-15-08934-f001:**
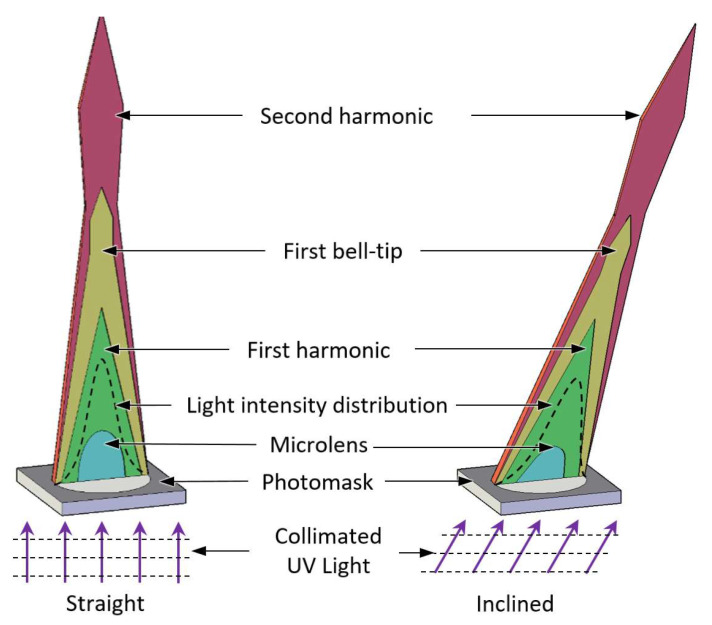
Four transition phases of the straight and inclined microneedles.

**Figure 2 materials-15-08934-f002:**
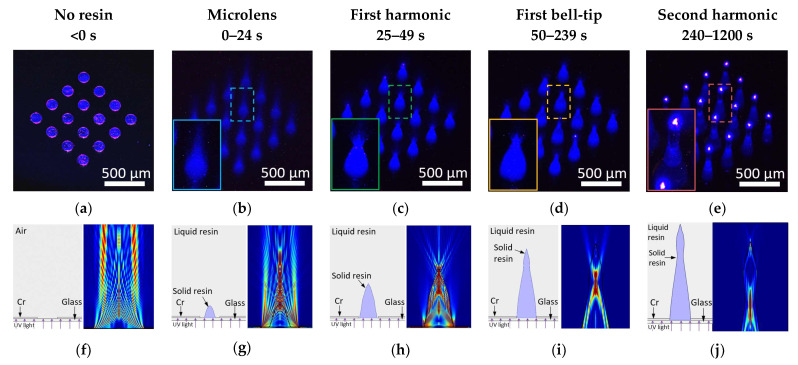
UV light propagation through the photomask before resin coating and during the four transition phases. (**a**) UV light propagation through the photomask patterns. (**b**) Forming microlens shape with focused light. (**c**) First harmonic shape as microneedle. (**d**) First bell-tip formed. (**e**) Second harmonic shape on top to the first harmonic shape. Simulation results of UV light propagation using COMSOL Multiphysics^®^ Software. (**f**) UV light propagation in air through a 150 µm slit. (**g**) UV light propagation in resin with presence of a microlens. (**h**) Diverging light propagation as light exited first harmonic. (**i**) UV light propagation during first bell-tip phase. (**j**) UV light propagation during second harmonic phase.

**Figure 3 materials-15-08934-f003:**
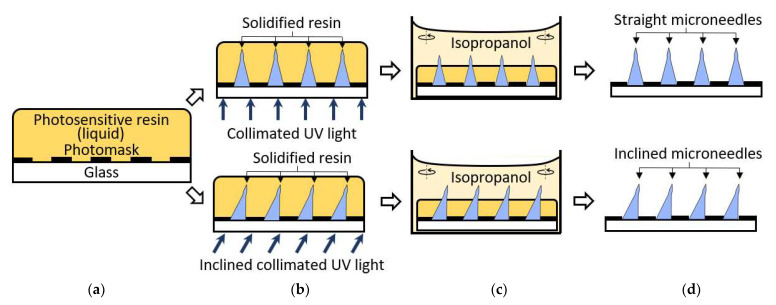
Straight and inclined microneedles fabrication process. (**a**) Photosensitive resin coated on the photomask. (**b**) Backside UV exposure. (**c**) Develop resin in isopropanol with slight agitation. (**d**) Microneedles array complete.

**Figure 4 materials-15-08934-f004:**
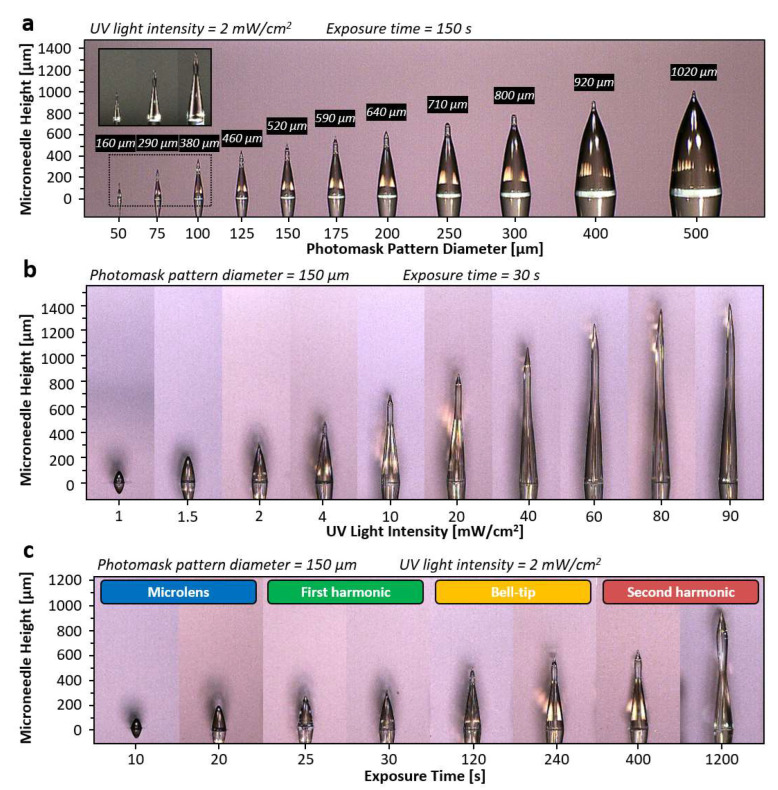
Microneedle characterization results based on the three parameters. (**a**) Photopattern diameter as the variable (50, 75, 100, 125, 150, 175, 200, 250, 300, 400, and 500 µm), microneedles height increases as diameter increases while microneedle phase progresses slower as diameter increases. Inset image show the enlarged view of the microneedles with diameter of 50, 75 and 100 µm. (**b**) UV light intensity as the variable (1, 1.5, 2, 4, 10, 20, 40, 60, 80, and 90 mW/cm^2^), microneedle forms higher tip to body or height to base ratio as the intensity increases given the same microneedle phase. (**c**) Exposure time as the variable (10, 20, 25, 30, 120, 240, 400, and 1200 s), microneedle progresses through microlens, first harmonic, bell-tip and second harmonic phases with increasing height as the exposure time increases.

**Figure 5 materials-15-08934-f005:**
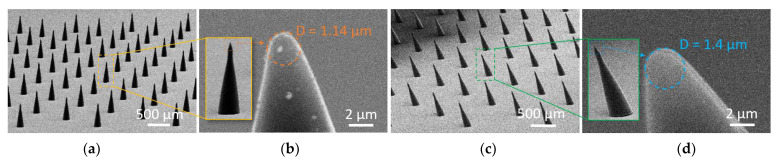
Straight and inclined first bell-tip microneedles arrays (180 units). (**a**) Straight microneedles array with diameter and height of 150 µm and 460 µm. (**b**) Zoomed-in view at the straight microneedle tip with tip diameter of 1.14 µm. (**c**) Inclined microneedles array with diameter, height, and inclined angle of 150 µm, 450 µm, and 20°. (**d**) Zoomed-in view at the inclined microneedle tip with tip diameter of 1.4 µm.

**Figure 6 materials-15-08934-f006:**
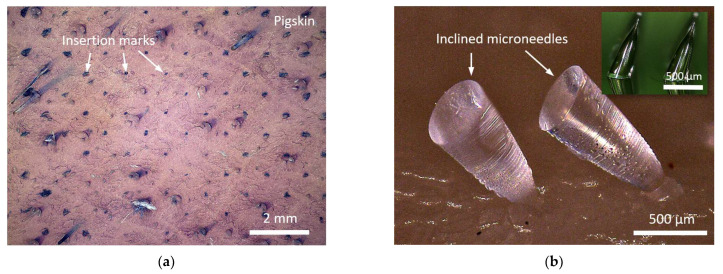
Skin insertion test results. (**a**) Pigskin inserted by 180 units PLA microneedles array with first bell-tip. (**b**) Successful tips insertion and needle body detachment using resin-based inclined microneedles array.

**Figure 7 materials-15-08934-f007:**
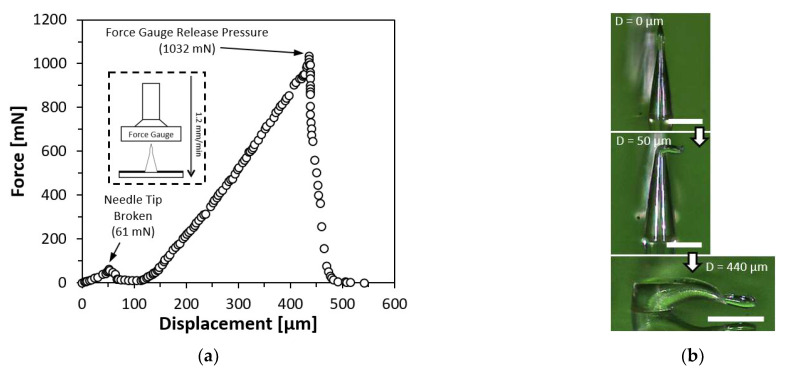
Force-displacement test result on a single PLA straight first bell-tip microneedle. (**a**) Force-displacement test result. (**b**) Compressed microneedle. Scale bar = 200 µm.

**Figure 8 materials-15-08934-f008:**
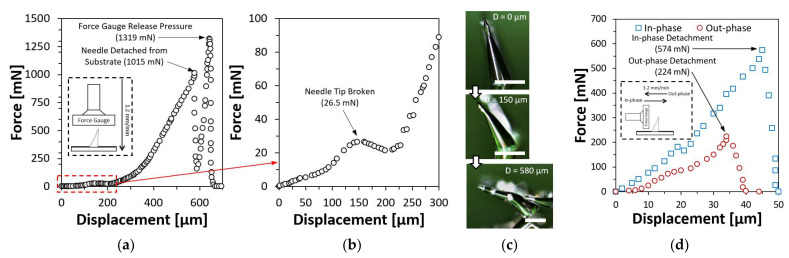
Force-displacement test results on a single resin-based inclined first bell-tip microneedle. (**a**) Force-displacement test result with vertical compression. (**b**) Zoomed-in at the first 300 µm displacement to visualize the force drop while the needle tip was broken. (**c**) Compressed microneedle. Scale bar = 200 µm. (**d**) Force-displacement test with in-phase and out-phase compression.

## Data Availability

Not applicable.

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
