# Peer review of "Experimental Validation of Diffraction Lithography for Fabrication of Solid Microneedles"

_materials, 2022, doi:10.3390/ma15248934_

Round 1
Reviewer 1 Report
Line 42: "owing to the outermost skin barrier" - explain with more detail here, what is happening.
57: the claim here is that solid drugs can be delivered, provide reference
118: reference FormLabs as manufacturer of surgical guide resin
Were the two resins that were used (Surgical Guide and Grey Resin) compared for UV transmissivity and does that affect the shapes that can be achieved? How does the selected resin itself affect the needle shape, if at all? If not, why not?
132: wavelength?
Fig 2. Time scale?
Paragraph Line 131 to 165- it is not clear what is simulation and what is not. Possibly the entire discussion is about the simulation but then words like 'observed' were used.
209: since the photomask area is large, did you verify light source intensity at all locations?
313: For skin insertion tests apparently a mold of PDMS was made of the resin needles. Then PLA is used to form new needles. However, earlier in the paper, surgical guide resin is used due to its biocompability. Why is PLA being used here? There seems to be a jump that the authors have not clearly explained.
Again, section 4.7 discusses a force study on PLA needles (that were straight) and section 4.8 shows a force study on resin based needles on an incline. Why were these specific studies chosen? why not a direct comparison of PLA to PLA or resin to resin or inclined to straight needles? None of the approach is explained.
Author Response
Thank you for your insightful feedback. We have compiled all reviewer comments and responses into a single PDF document.

Reviewer 2 Report
Dear respected authors,
In this paper, in order to create microneedles, you researched the liquid-to-solid phase transition using UV diffraction. You analyzed how the microneedles' morphology changed in response to exposure to UV light parameters.
But I do have a few comments on your article as follows:
1- First of all, the distribution of cited sources is insufficient. Not only are some general important statements supported by a single citation, but you also did not mention their reference for some statements. For example, on Line 75: Which group was recently present a low-cost, quick UV lithography technology for microneedle production? Or Line 82.
2- Following the presentation of Kim et al.'s work at the conference, is there another group working on this method, or do they publish additional articles?
3- You mentioned that you've used COMSOL to simulate your proposed method on page 3. Please present the structure you described as well as the simulation's findings.
4- Please provide the technical specifications for the photosensitive resin that was referred to. Furthermore, what are the glass material and structural properties used as the substrate? Was the resin adhering well to the glass? What was the point of not washing out the photomask? Including a photo of your lighting setup is helpful. Is backlighting or undercutting something you need to think about when the density or duration of UV exposure increases? Is there a correlation between photomask thickness and these values?
5- Is it possible to formulate the relation between the characteristics of the needles and the three parameters discussed in the article?
6- In large-scale fabrication, have you investigated the distribution of needle structure parameters?
7- Page 9: Please explain the dissociation of the needle body from the glass substrate.
Thank you
Author Response

(The authors gave the same response as above.)

Reviewer 3 Report
The manuscript mainly reflects the relationship between the applied UV radiation and the morphology of the microneedles by SLA-3D using commercial liquid-state photosensitive resin. The correlation of each light diffraction parameter connected to the characteristics of the microneedle is examined to elucidate the four different phases observed during the photosensitive resin's liquid to solid transition. Some meaningful conclusions were obtained. There are still some problems in the manuscript, which need to be further modified.
Comment 1: The components of photosensitive resin are various, and different components have different chemical and physical properties. Moreover, most kinds of photosensitive resins have certain toxicity to human body and irritation to skin. Has the biosafety of materials been considered or biosafety experiments been conducted in this experiment? Although the experiment uses large guide resin, we still do not know its composition and biosafety coefficient. Can we consider quoting others' research in this regard?
Comment 2: In this experiment, the mechanical test of the microneedle was carried out, but in fact, with the size of the microneedle, it does not require too much force to penetrate the human skin. Human is an organism with various chemicals. Have you considered the solubility of resin in human skin or cited other people's research in this field.
Author Response

(The authors gave the same response as above.)

Reviewer 4 Report
The manuscript reports the recent progress in the process control for the fabrication of microneedles using the diffraction lithography method. The article is very well presented, the authors explained the potential as well as the challenges in this research. In this reported work, the authors studied the influence of the key parameters including the pattern size, the UV light intensity, the exposure time, the resin refractive index, etc. The exposure was simulated to help analyze the resin solidification process and the final microneedle profile. The simulation results have good agreement with the experimental data. The referee also noticed that the authors achieved much better control over the microneedle profile, compared to what they reported two years ago (ref.41 in the manuscript).
Considering that the topic is relevant to the scientific community and matches the journal's purpose, the referee suggests the submitted manuscript can be published in the journal Materials as it is.
The comments below are just out of the referee’s curiosity, not affecting the decision of the article itself:
1) The authors reported the possibility of fabricating the ‘multi-leg’ microneedles with the same approach (DOI: 10.1109/NEMS54180.2022.9791201). However, the microneedles fabricated with the reported method are all solid ones, is there a possibility of producing tube-like microneedles (enabling fluid injection)?
2) The microneedle arrays can also potentially be used as electrodes in brain-machine-interface (BMI) if they become conductive. Can the microneedles be ‘printed’ with conductive photosensitive resins?
3) The author reported the four different types of cross-link types, which might have different micro-fiber orientations. Have the authors ever considered examining the fiber orientations with methods like the X-ray dark-field imaging (small angle scattering) method?

Author Response

(The authors gave the same response as above.)
